# Pollen Proteases Play Multiple Roles in Allergic Disorders

**DOI:** 10.3390/ijms21103578

**Published:** 2020-05-19

**Authors:** Ricardo Gaspar, Mafalda Ramos de Matos, Luísa Cortes, Isabel Nunes-Correia, Ana Todo-Bom, Euclides Pires, Paula Veríssimo

**Affiliations:** 1CNC-Center for Neuroscience and Cell Biology, University of Coimbra, 3004-504 Coimbra, Portugal; ricardo.gaspar@inl.int (R.G.); mafaldaRmatos@msn.com (M.R.d.M.); corteluisa@gmail.com (L.C.); inunescorreia@gmail.com (I.N.-C.); euclides.pires@gmail.com (E.P.); 2Microscopy Unit Center for Neuroscience and Cell Biology, University of Coimbra, 3004-504 Coimbra, Portugal; 3Immunoallergology Service, Coimbra University Hospital, 3004-504 Coimbra, Portugal; atodobom@fmed.uc.pt; 4Faculty of Medicine, University of Coimbra, 3000-504 Coimbra, Portugal; 5Department of Life Sciences, University of Coimbra, Calçada Martim de Freitas, 3000-456 Coimbra, Portugal

**Keywords:** allergy, pollen proteases, transepithelial permeability, IL-6, IL-8 and PAR-2

## Abstract

Allergic diseases are a major health concern worldwide. Pollens are important triggers for allergic rhinitis, conjunctivitis and asthma. Proteases released upon pollen grain hydration appear to play a major role in the typical immunological and inflammatory responses that occur in patients with allergic disorders. In this study, we aimed to identify specific proteolytic activity in a set of pollens with diverse allergenic potential. Diffusates from *Chenopodium album*, *Plantago lanceolata* and *Eucalyptus globulus* were added to a confluent monolayer of Calu-3 cells grown in an air-liquid interface system. We identified serine proteases and metalloproteinases in all pollen diffusates investigated. Proteases found in these pollen diffusates were shown to compromise the integrity of the lung epithelial barrier by disrupting transmembrane adhesion proteins E-cadherin, claudin-1 and Occludin, as well as, the cytosolic complex zonula occludens-1 (ZO-1) resulting in a time-dependent increase in transepithelial permeability. Tight junction disruption and increased transepithelial permeability facilitates allergen exposure to epithelial sub-layers contributing to the sensitization to a wide range of allergens. These pollen extracts also induced an increase in the release of interleukin 6 (IL-6) and interleukin 8 (IL-8) cytokines measured by flow cytometry possibly as a result of the activation of protease-activated receptors 2 (PAR-2).

## 1. Introduction

Pollen allergy has a clinical impact all over the world. Allergic diseases, which include seasonal rhinitis and asthma, are recognized as inflammatory disorders of the airway mucosa typically driven by a immunoglobulin E (IgE) mediated response [1]. During recent decades, the prevalence and severity of respiratory allergic reactions induced by pollens have increased [2,3,4,5,6,7]. The majority of patients display adverse reactions upon contact with pollen allergens fostering the need for further studies of different types of pollens. The most clinically relevant allergenic pollens in Europe are grass pollen followed by *Betula alba*, *Alnus incana*, *Corylus avellana*, *Platanus vulgaris*, *Cupressus sempervirens*, *Olea europaea*, *Artemisia vulgaris*, *Ambrosia artemisiifolia* and *Parietaria Judaica* [8]. We previously showed that the proteolytic activity of pollen diffusates from *Olea europea*, *Dactylis glomerata*, *Cupressus sempervirens* and *Pinus sylvestris* compromise the airway epithelial barrier, an effect that is reversed by serine proteases specific inhibitors [9].

This study focused on three pollen species that have different IgE mediated allergenic capacities, *Chenopodium album* (moderate allergenicity), *Plantago lanceolata* (moderate allergenicity) and *Eucalyptus globulus* (low allergenicity) [10]. These species are widespread throughout southern Europe and have pollination peaks that overlap [11]. *Chenopodium album* as well as *Salsola kali*, both from the *Chenopodiaceae-Amaranthaceae* family, are perennial plants of the *Chenopodiaceae* family and have been reported to be responsible for hay fever in dry areas [12,13]. Pollinosis and allergic sensitization to the pollen of this weed have been reported in different European countries, mostly in Spain and also in North America, Australia and in Middle Eastern countries where it is recognized as one major trigger for rhinitis and asthma [14,15,16,17]. The allergenicity of *Chenopodium album* has been reported among atopic patients. The antigenicity of the protein fractions was confirmed by the interaction of patients’ sera on the immunoblot with IgE antibodies [18]. The *Plantago* genus of the *Plantaginaceae* family includes approximately 250 species. These species grow in humid meadows and on roadsides, invade lawns, spread steadily and are considered one of the most important dicotyledons that cause allergic diseases in Europe [19]. Weed pollen allergic patients are often polysensitized. Based on molecular diagnosis it was possible to have an accurate identification of patients sensitized to *Plantago lanceolata* that recognized Pla l 1 besides other cross reactive allergens [20,21]. *Eucalyptus* belongs to the *Myrtaceae* family and the allergens present in *Eucalyptus globulus* are considered to have low allergenic potential [10].

Pollen grains carry a variety of molecules that are released upon hydration, including nicotinamide adenine dinucleotide phosphate reduced form (NADPH) oxidases, substances that act as inflammatory mediators, such as prostaglandin E2 (PGE2) and leukotriene B4 (LTB4), and many proteins, including proteases [22,23]. Proteases have been involved in airway epithelium barrier degradation by increasing epithelial permeability [7,9]. This activity ultimately facilitates the activation of the immune system by contact with allergens; this process can initiate or exacerbate allergic responses, specific IgE production and perpetuate type 2 inflammation [22,24,25,26,27,28]. 

The loss of epithelial barrier integrity may also occur via the activation of protease-activated receptors 2 (PAR-2), which are widely expressed on epithelial cells, endothelial cells, airway smooth muscle and fibroblasts [29]. Allergens that possess proteolytic activity can activate PARs on the surface of cells promoting the release of chemokines, cytokines and growth factors leading to the activation of epithelial cells and other immune cells. PAR-2 is widely expressed on the apical surfaces of Calu-3 epithelial cells [30]. All of the classical second messengers that are produced subsequent to PAR-2 activation, namely Ca^2+^, cyclic adenosine monophosphate (cAMP), guanine nucleotide-binding proteins (G proteins) and protein kinase C (PKC) have been reported to influence the properties of tight junction complexes. PAR-2 activation appears to promote the opening of calcium channels and the influx of calcium into the cytoplasm [24,31,32]. The proteolytic activation of PAR-2 could indirectly act in the degradation of the epithelium through the activation of internal inflammatory signaling cascades [24,29]. 

This study aimed to characterize the major pollen proteases of *C. album, P. lanceolata* and *E. globulus*, to identify possible targets that compromise lung epithelial barrier integrity and to explore the effect of proteases present in these pollen diffusates on the inflammatory response by the quantification of cytokine release. 

## 2. Results

### 2.1. Hydration of Pollen Grains from Chenopodium Album, Plantago Lanceolata and Eucalyptus Globulus Leads to the Release of Serine Proteases and Metalloproteinases

Pollen diffusates were obtained by hydration in 50 mM Tris-HCl pH 7.4 (~20 mg of pollen/ml) for 2 h at room temperature. Different pollens exhibited distinct abilities to release their solutes once hydrated. Specifically, the quantity of total protein released by *E. globulus*, which had a low allergenic potential and a larger pollen grain size, was superior (0.75 ± 0.18 mg/ml) in comparison to pollen extracts from *C. album* and *P. lanceolata,* (0.26 ± 0.06 mg/ml and 0.18 ± 0.03 mg/ml, respectively), which had moderate allergenic potentials and small pollen grain dimensions. Pollen grains were hydrated in vitro to mimic the process that occurs in the respiratory airways when an individual is exposed to pollen inhalation. We used the same initial proportion of pollen mass per volume of buffer to prepare the three diffusates.

In order to obtain a rapid proteolytic profile, each pollen diffusate was analyzed by two-dimensional gelatin zymography. All pollen diffusates contained gelatin degrading proteases which were of high molecular weight and acidic pI (Appendix A). Enzymatic assays using 7-amino-4-methylcoumarin (AMC) synthetic substrates and class-specific protease inhibitors were performed to characterize the proteolytic activity present in the pollen diffusates. All pollen diffusates exhibited proteolytic activity against a variety of AMC-coupled substrates (Figure 1A), primarily AMC linked to phenylalanine (Phe-AMC), AMC linked to leucine (Leu-AMC) and AMC linked to methionine (Met-AMC). *E. globulus* exhibited higher specific enzymatic activity than *C. album* and *P. lanceolata* pollen diffusates, which were quite similar to each other. As Phe-AMC was the preferred substrate for all pollen diffusates, this substrate was used to evaluate pollen protease specificity. 

The high inhibition of 4-(2-aminoethyl) benzene-sulfonyl fluoride hydrochloride (AEBSF), a serine protease inhibitor, and ethylenediaminetetraacetic acid (EDTA) suggests the existence of a preferential serine and metalloproteinase activity in the three pollen species (Figure 1B). We observed a statistically significant inhibition of *C. album* by tosyl phenylalanyl chloromethyl ketone (TPCK) which is a class-specific inhibitor of serine-type chymotrypsin-like proteases. *P. lanceolata* was also significantly inhibited by tosyl-L-lysine chloromethyl ketone (TLCK), class-specific inhibitor of serine-type trypsin-like proteases. Both *C. album* and *E. globulus* pollen diffusates also contain cysteine activity as indicated by partial inhibition by the proteinase inhibitor E-64 (Figure 1B). 

### 2.2. Pollen Proteases Increase Transepithelial Permeability by Disrupting Protein Intercellular Complexes

The precise mechanism by which allergens overcome the epithelial barrier has not yet been fully understood. Pollen proteases have been suggested to cleave intercellular protein junctions, increasing transepithelial permeability promoting contact with antigen-presenting cells (APCs) in subepithelial layers ultimately leading to allergic responses [9,26,27,28]. The effect of pollen proteases on the paracellular integrity of the epithelial barrier was evaluated using Calu-3 cells grown in air-liquid interface culture [33]. The Calu-3 cell line is one of the airway epithelial cell lines that form functional tight junctions in vitro, resulting in high transepithelial resistance [34]. This system provides a functional model for airway epithelial barrier studies [33]. We observed that all pollen diffusates induced an increase in transepithelial permeability, in a time-dependent manner (Figure 2A). *C. album* induced a statistically significant increase in the transepithelial flux of rhodamine B isothiocyanate dextran conjugated (RITC-dextran). *P. lanceolata* and *E. globulus* have a weaker effect but still significant. 

To determine whether the observed increase in transepithelial permeability was related to the presence of proteases, pollen diffusates were pre-treated with 1 mM AEBSF and then incubated with Calu-3 cells for 12 h. AEBSF, serine protease inhibitor, completely inhibited the effects of *P. lanceolata* and *E. globulus* pollen diffusates while for *C. album* the inhibition was partial (Figure 2B). 

To understand the mechanism that leads to an increase in transepithelial permeability and cell detachment, potential protein targets for disruption [28,35,36] were identified using immunoblot (Figure 3) and immunocytochemistry (Appendix A). All pollen diffusates caused extensive and significant degradation of E-cadherin, claudin-1, occludin and the cytosolic protein zonula occludens-1 (ZO-1) (Figure 3). *C. album* had a more drastic effect when compared to *P. lanceolata* and *E. globulus.* The protease degradation was inhibited after treatment with AEBSF. Regarding *P. lanceolata* and *E. globulus*, there was complete inhibition while only partial inhibition was observed for *C. album* (Figure 3). These results suggest that the presence of other non-serine proteases was capable of inducing the degradation of intercellular junction proteins.

The effect of pollen diffusates on intercellular protein complexes were further evaluated by immunocytochemistry. Microscopy detection of disrupted protein junctions could be easily observed by evident interruptions in the continuous ring at the epithelial cell apices, caused by the disassembly of macromolecular protein complexes. As expected, all pollen diffusates induced interruptions in the epithelial cell apices when labeled for E-cadherin, claudin-1, occludin and ZO-1 (Appendix A). Noteworthy, is the fact that *C. album* was used in this assay at a 1:20 dilution as this pollen diffusate induced a high degree of cell detachment. 

### 2.3. Pollen Proteases may Activate PAR-2 and Induce IL6 and IL8 Release

The variations in [Ca^2+^]_i_ at the single-cell level were analyzed after exposure to different pollen diffusates. Interestingly, Calu-3 cells exhibited an increase in [Ca^2+^]_i_ in response to pollen diffusates from *C. album* but not after exposure to denatured *C. album* extracts (95 °C for 30 min), suggesting a possible role of proteases in PAR-2 activation (Figure 4B, C). It is worth noting that the effect of *C. album* diffusates was so drastic that the cells were unable to return to basal conditions after exposure to this pollen extract. The effect of *E. globulus* pollen diffusates on [Ca^2+^]_i_ levels was less prominent than that of *C. album*, although this effect remained significant (Figure 4C). *P. lanceolata* diffusates were incapable of increasing [Ca^2+^]_i_ levels possibly due to the low amount of proteases present in this pollen extract.

Flow cytometry was used to quantify the interleukin 6 (IL-6) and interleukin 8 (IL-8) released by Calu-3 cells after a 6 h exposure to pollen diffusates or denatured pollen diffusates (95 °C for 30 min). All pollen diffusates induced the release of IL-6 and IL-8. Moreover, this effect was not observed with denatured diffusates supporting the relevance of pollen proteases for the initial steps of the allergic inflammatory response (Figure 5). 

## 3. Discussion

Pollen proteases are important triggers of allergic disorders, containing a large number of allergens and proteolytic enzymes. Previous studies demonstrated that the most important sensitizing pollens in southern Europe (i.e., *Olea europaea, Dactylis glomerata* and *Parietaria judaica*) release proteases that cause the detachment of human epithelial cells by degrading intercellular adhesion proteins and facilitate allergen delivery across the epithelium [9,27]. However, little information is available concerning other relevant allergenic pollens, such as, *C. album*, *P. lanceolata* and *E. globulus,* which are common causes of pollinosis in the Mediterranean area. 

The healthy bronchial epithelium is an impermeable barrier that provides resistance against the paracellular flow of macromolecules and infectious agents due to specialized cell junctions [34]. Many recent studies focused on the identification and characterization of the proteolytic activity present in allergic sources, such as dust mites, fungus and pollens [6,24,26,37]. This enzymatic activity has been suggested to be responsible for triggering allergic responses as a result of altering the integrity of the lung epithelium which allows allergens to gain access to and contact APCs to induce IgE production [26]. After inhaled, pollen is hydrated on the surface of the respiratory epithelium, which leads to the rapid release of its contents, including allergens, proteases, NADH oxidases, lipoproteins, polysaccharides, lipids and phenolics [9,26]. High quantities of pollen solutes can be concentrated in the airway epithelium mucosa, depending on the type of pollen, the individual’s intensity of exposure to the pollen, the geographic location and time of year [9,26]. In this work, the concentration of protein released by an equal mass of pollen ranged from 0.18 mg/ml to 0.75 mg/ml. This variation in protein concentration among the analyzed pollens was taken into account in all experiments. The characterization of pollen proteolytic activity revealed a prevalence of serine and metalloproteinases in all three pollen diffusates. Moreover, cysteine activity was observed in *C. album* and *E. globulus* pollen diffusates. These hydrolytic specificities are observed for the majority of aeroallergen proteases [6].

Epithelial damage has been recognized as key in driving airway inflammation and remodeling. The integrity of the barrier formed by the epithelium that lines the airways is dependent upon the continuity of the superficial columnar cell layer and the effectiveness of the adherent and tight junctions [28]. Tight junctions located on the apical surface of epithelial cells form a diffusion barrier that regulates the flux of ions and hydrophilic molecules through the paracellular pathway [35,36]. Our study suggests that pollen proteases alter the barrier function of polarized epithelia, which was demonstrated both by significant increases in transepithelial permeability and by the degradation of intercellular protein complexes. Immunoblotting and immunofluorescence studies revealed a dramatic disruption of E-cadherin, claudin-1, occludin and the cytosolic complex ZO-1, suggesting that the observed increase in permeability may be due in part to the proteolysis of these proteins. Of the studied proteins, E-cadherin, which has long been known to contribute to the assembly of other specialized cell-cell junctions [38], was the most extensively degraded by pollen diffusates. This effect was most drastic for *C. album*, suggesting a link between cysteine and serine proteolytic activity and intercellular protein complex disruption and cell detachment. The effects of pollen proteases on cell detachment and barrier impairment may cause a localized increase in epithelial permeability, which may result in a higher probability of pollen allergen detection by antigen-presenting dendritic cells and the exposure of respiratory nervous endings [39].

Epithelial cells typically express multiple protease-activated receptors, and physiological responses depend on cross talk between these different receptors [40]. Studies using mite allergens, such as *Dermatophagoides pteronyssinus*, that possess endogenous proteases such as cysteine (Der p 1) and serine proteases (Der p 3, 6 and 9), revealed a capacity to activate PARs leading to inflammation through epithelial cell detachment with IgE and cytokine production [41,42,43,44]. In addition, several allergens are able to trigger a type 2 immune response in the absence of specific IgE antibodies, by activating airway epithelial cells via PAR [45].

Recent studies demonstrated that the activation of PAR-2 receptors can induce the release of matrix metalloproteinase-9 (MMP-9), granulocyte-monocyte colony-stimulating factor, eotaxin, PGE_2_, IL-6 and IL-8. These findings suggest that PAR activation may mediate the stimulation of airway epithelial cells and promote the development of type 2 inflammation associated to allergic disorders [42,43,46,47,48]. Moreover, previous studies demonstrated that the hydrolytic activation of PAR-2 leads to phospholipase Cβ activation via the G_q/11_α protein, which is a common pathway for G protein-coupled receptors. The activation of this pathway induces the production of inositol triphosphate, followed by Ca^2+^ mobilization [49,50]. 

In the present study, stimulation with *C. album* and *E. globulus* induced an increase in [Ca^2+^]_i_ in Calu-3 cells. Although both pollen diffusates induced an [Ca^2+^]_i,_ increase, the response was different for each pollen diffusate_._ When exposed to *C. album*, the effect on [Ca^2+^]_i_ was higher, and the cells were unable to return to basal conditions suggesting that *C. album* induced cell detachment even after exposure for short periods of time. 

The effect of aeroallergen proteases, which act via PAR-2, is also implicated in the regulation of proinflammatory cytokine release by epithelial cells, such as IL-6 and IL-8, during nonallergic inflammatory responses [24,42,51]. Although IL-6 and IL-8 release can occur via a mechanism that is independent of Ca^2+^ mobilization and PAR activation, our results point toward a pathway in which serine and/or cysteine pollen proteases activate PAR-2 in the airways and induce IL-6 and IL-8 release. These results are consistent with those obtained with house dust mite and fungi extracts, which contain serine proteases and have been shown to directly induce airway inflammation characterized by IL-6 and IL-8 release in the airway epithelium [24,52,53,54]. This investigation shows for the first time that pollen proteases from *C. album*, *P. lanceolata* and *E. globulus* induced proinflammatory IL-6 and IL-8 cytokine production by Calu-3 human airway epithelial cells, likely mediated by PAR-2 activation. These effects were dependent on the proteolytic activity released by pollen grains. 

Based on our results, we propose that the protease activity released by pollen grains compromise the barrier integrity of the lung epithelium, thereby facilitating the passage of allergens through the epithelium leading to sensitization and the initiation of an inflammatory response. This study provides additional pathophysiological insights into common allergic disorders, which may prove useful in the search for possible therapeutic targets for the treatment of respiratory diseases.

## 4. Materials and Methods 

### 4.1. Preparation of Pollen Diffusates

The protein extracts from *C. album*, *P. lanceolata* and *E. globulus* (Allergon, Ängelhom, Sweden) were prepared by stirring 400 mg of pollen grains in 20 ml of 50 mM Tris-HCl buffer pH 7.4 for 2 h at room temperature. The homogenized samples were centrifuged at 12,000 g for 10 min at 4 °C. The supernatant was collected, filtered using a 0.45 µm filter (Orange Scientific, Braine-l’Alleud, Belgium) and stored in aliquots at −20 °C. Protein concentration was determined using the Bio-Rad Protein Assay with bovine serum albumin (BSA, Bio-Rad, Hercules, California, CA, USA) as a standard.

### 4.2. Enzymatic Activity Assay

The proteolytic activity of the pollen diffusates was determined using peptide substrates coupled to 7-amino-4-methylcoumarin (AMC). For this enzymatic assay, 200 µl of pollen diffusates (protein concentration ranging between 40–160 μg/ml) was incubated with 100 µM of fluorescent substrate. The hydrolysis and release of AMC was monitored every 15 s for 20 min at 37 °C using a SpectraMAX-GeminiEM fluorimeter (Molecular Devices, Sunnyvale, CA, USA).

The specific protease inhibitors required a 20 min pre-incubation period with the pollen diffusates before monitoring enzymatic activity. The enzymatic activity in this assay was assessed using the preferential AMC-substrate of each pollen diffusate in the same manner as described previously. 

### 4.3. Cell Culture

Calu-3(ATCC ® HTB-55™) cells were purchased from the American Type Culture Collection (Manassas, Virginia, VA, USA). Calu-3 cells were grown using an air-liquid interface culture (AIC) in which epithelium cell polarization occurs [33]. The cells were cultured in Dulbecco’s modified Eagle´s medium (Sigma, St. Louis, MO, USA) supplemented with 10% heat-inactivated fetal bovine serum (FBS) (Gibco, Barcelona, Spain), 100 U/ml of penicillin and 100 µg/ml of streptomycin (Gibco). Cell cultures were grown at 37 °C in a humidified atmosphere with 5% CO_2_/95% air. 

### 4.4. Transepithelial Permeability Measurement

Calu-3 cells were seeded on transwell filter inserts with a 0.4 μm pore size and placed into 12-well tissue culture plates (Corning, Bath, UK). The cells were grown using an air-liquid interface culture [9]. The chambers, termed inserts, were washed daily with PBS and the culture medium was renewed. The cells were maintained in culture under these conditions until they reached confluency. Cells were washed with PBS and incubated for 2 h with serum-free culture medium. Then, each pollen diffusate was added to the apical chamber for different incubation times ranging from 1 h to 12 h at 37 °C. Calu-3 cells were also exposed to pollen diffusates that were pre-treated for 20 min with a 1 mM solution of the 4-(2-aminoethyl) benzenesulfonyl fluoride (AEBSF) inhibitor. 

Time points 1, 3, 6 and 12 h were chosen for the stimulation period. 50 µl of 100 μM rhodamine isothiocyanate (RITC)-dextran (70-kDa) (Sigma) was added to the apical insert of each chamber of the transwell plate at a final concentration of 10 μM. The fluorescence of the basolateral medium samples was quantified using a SpectraMAX-GeminiEM fluorimeter (Molecular Devices, Sunnyvale, CA, USA) with an excitation wavelength of 530 ± 25 nm and an emission wavelength of 590 ± 35 nm (SoftMax Pro v5 software, Molecular Devices, Sunnyvale, CA, USA). Epithelial permeability was calculated as the percentage of RITC-Dextran 70S present in the medium from the bottom chamber in comparison to the amount added to the top chamber solution per filter surface area (cm^2^) per unit of time (s).

### 4.5. Electrophoresis and Immunoblotting 

To evaluate the integrity of tight junction integral proteins, cells were washed with serum-free medium and incubated for 6 h with each pollen diffusate. Cells were also exposed to pollen diffusates that were pre-treated with 1 mM AEBSF. After incubation, Calu-3 cells were washed twice with PBS containing protease inhibitors (1 μg/ml of chymostatin, leupeptin, antipain and pepstatin (CLAP); 1 mM DTT; and 0.1 mM PMSF), scraped into tubes and centrifuged (14,000 rpm for 5 min at 4 °C). Denaturing sample buffer (100 mM Tris-Bicine, 6 M urea, 4% SDS, 4% 2-β-mercaptoethanol, and 0.002% bromophenol blue) was added to the cell extracts which were then sonicated and newly centrifuged (14,000 rpm for 5 min at 4 °C). The supernatants were collected. After SDS-PAGE the proteins were transferred electrophoretically to PVDF membranes (Millipore) and immunoblotting was performed according to a standard protocol [55]. The following primary antibodies were used: mouse monoclonal anti–E-cadherin (1:500), rabbit monoclonal anti-claudin (1:250), mouse polyclonal anti-occludin (1:250) and mouse polyclonal anti-ZO-1 (1:250) (Zymed Laboratories, Barcelona, Spain). The secondary antibodies were alkaline phosphatase-conjugated anti-rabbit or anti-mouse IgG (GE Healthcare, Buckinghamshire, UK). After incubating the membrane with ECF reagent, detection was performed via enhanced chemifluorescence (GE Healthcare, Buckinghamshire, UK). The immunoblots were normalized to β-actin (1:5000) (mouse monoclonal anti-actin, Sigma, St. Louis, MO, USA).

### 4.6. Single-Cell Calcium Imaging

Calu-3 cells were grown on 12 mm glass coverslips and maintained in culture until confluency. Cells were loaded with 5 µM Fura-2/AM (Molecular Probes, UK), 0.1% fatty acid-free BSA and 0.02% pluronic acid F-127 (Molecular Probes) in Krebs solution with calcium (132 mM NaCl, 4 mM KCl, 1.4 mM MgCl_2_, 1 mM CaCl_2_, 6 mM glucose, and 10 mM HEPES, pH 7.4) for 60 min in an incubator with 5% CO_2_/95 % air at 37 °C. To achieve complete hydrolysis of the probe, cells were then incubated in Krebs solution for 10 min. The glass coverslip was mounted on an RC-20 chamber in a PH3 platform (Warner Instruments, Hamden, CT, USA) on the stage of an Axiovert 200 inverted fluorescence microscope (Carl Zeiss, Jena, Germany). The microscope was equipped with a lambda DG4 apparatus that used 340/20 nm and 380/20 nm excitation filters and a 510/30 nm band-pass emission filter (Chrona TechCorporation, Vermont, VT, USA). Images were acquired using a Plan Neofenar 40×/0.75 objective (Carl Zeiss, Oberkochen, Germany) and a CoolSNAP digital camera (Roper Scientific, Trenton, NJ, USA). Variations in [Ca^2+^]_i_ were evaluated by quantifying the ratio of the fluorescence emitted by Fura-2 at 510 nm after excitation at 340 and 380 nm. The acquired values were processed using the MetaFluor software (Universal Imaging Corp., West Chester, PA, USA). As PAR-2 is selectively activated by trypsin and trypsin-like enzymes, the effect of a trypsin solution (0.25%) on [Ca^2+^]_i_ was used as a positive control.

### 4.7. Cytokine Quantification by Flow Cytometry

Cytokine levels were measured using the Human Basic FlowCytomix Multiplex kit (Bender MedSystem GmbH, Vienna, Austria) according to the manufacturer’s instructions. The assay is based on a mixture of beads of different sizes that are coated with capture antibodies specific for each cytokine to be analyzed. A biotin-conjugated secondary antibody mixture was used to detect the cytokine of interest. After washing to remove unbound antibody, a streptavidin-phycoerythrin reagent was added. After removal of unbound material by washing, the bead suspension was analyzed using a FACSCalibur flow cytometer (BD Biosciences, San Jose, CA, USA). Bead size and bead fluorescence identify the type of cytokine, and phycoerythrin fluorescence was used as a measure of cytokine concentration based on standards of known concentrations. The results were analyzed using the FlowCytomix Pro 2.4 Software (Bender MedSystem GMbH) and expressed in pg/ml for each cytokine. 

### 4.8. Statistical Analysis

The results are presented as the mean ± SEM. Statistical analysis was performed using one-way ANOVA followed by Dunnett’s post-test (inhibition assays) or two-way ANOVA followed by Bonferroni´s post-test (permeability assays, immunoblot studies, single-cell calcium imaging and cytometric bead array studies). Differences were considered significant when *p*-values were <0.05. 

## 5. Conclusions

This work demonstrated for the first time that pollen diffusates from *C. album*, *P. lanceolata* and *E. globulus* release proteases with a significant in vitro effect on the paracellular permeability of epithelial cells and induced major degradation of cell junction proteins: E-cadherin, claudin-1, occludin and ZO-1. Moreover, our data suggests that the loss of effectiveness of the epithelium barrier may also occur via activation of PAR-2 followed by an immune response characterized by an increase in the release of IL-6 and IL-8.

## Figures and Tables

**Figure 1 ijms-21-03578-f001:**
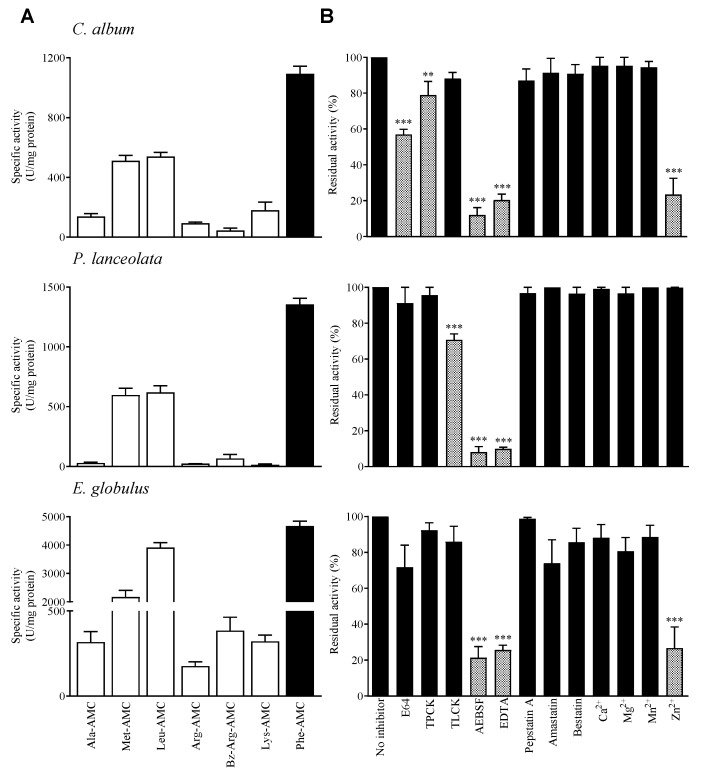
Specificity of pollen proteases from *Chenopodium album*, *Plantago lanceolata* and *Eucalyptus globulus* diffusates. (**A**) Substrate specificity of pollen proteolytic activity. The extracts were incubated for 20 min at 37 °C with 100 µM peptide-AMC. The black bars indicate the preferred substrate for each pollen diffusate, which was also the substrate used in the inhibition experiments. (**B**) Effect of class-specific inhibitors on the proteolytic activity of the pollen diffusates. The results represent the percentage of residual activity in comparison to the control condition (without inhibitor). The grey bars indicate statistically significant inhibition results. Dunnett’s post-test: *** *p* < 0.001; ** *p* < 0.01(*n* = 3). 1 unit of activity (U) = pmol AMC released/min/mg of protein.

**Figure 2 ijms-21-03578-f002:**
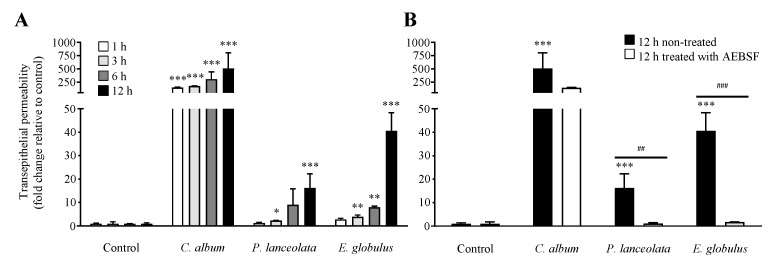
Effect of pollen proteases from *Chenopodium album* (0.26 ± 0.06 mg/ml), *Plantago lanceolata* (0.18 ± 0.03 mg/ml) and *Eucalyptus globulus* (0.75 ± 0.18 mg/ml) on the paracellular permeability of epithelial cells. (**A**) Time-course effect of pollen diffusates on Calu-3 cells. (**B**) Effect of untreated pollen diffusates (black bars) and diffusates treated with 1 mM AEBSF (white bars) on the transepithelial permeability of Calu-3 cells. Cells incubated with culture medium were used as a control for this assay. Bonferroni post-test: *** *p* < 0.001; ** *p* < 0.01; * *p* < 0.05 (*n* = 3) in comparison to the control condition, and *^###^ p* < 0.001; *^##^ p* < 0.01 (*n* = 3) for comparisons between treated and untreated conditions.

**Figure 3 ijms-21-03578-f003:**
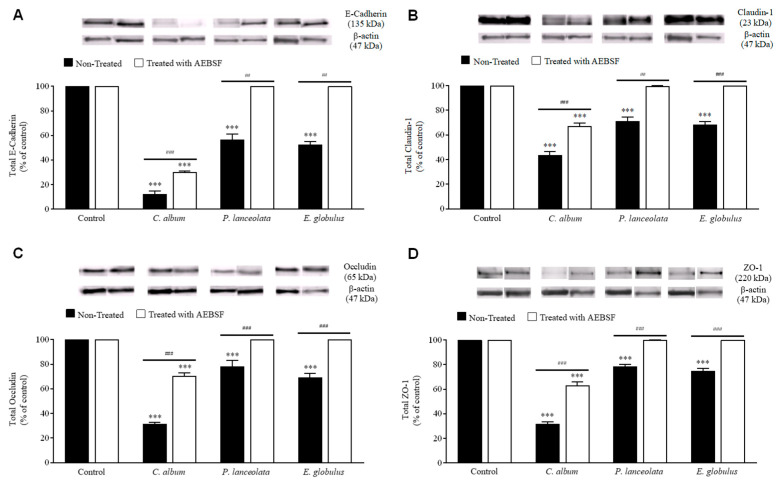
Effect of pollen diffusates on intercellular junction integrity. Calu-3 cells were exposed to pollen diffusates from *Chenopodium album* (0.26 ± 0.06 mg/ml), *Plantago lanceolata* (0.18 ± 0.03 mg/ml) and *Eucalyptus globulus* (0.75 ± 0.18 mg/ml) (black bars) for 6 h. The cells were also incubated with pollen diffusates that were pre-treated with 1 mM AEBSF (white bars). The degradation of the transmembrane proteins E-cadherin (**A**) claudin-1 (**B**) occludin (**C**) and the cytosolic complex ZO-1 (**D**) was analyzed by immunoblotting. Cells incubated with culture medium were used as a control. The data were normalized and expressed as a percentage relative to each control. Representative blots are shown for each stimulus. Bonferroni post-test: *** *p* < 0.001 (*n* = 5) in comparison to the control condition, and *^###^ p* < 0.001; *^##^ p* < 0.01 (*n* = 5) for comparisons between treated and untreated conditions.

**Figure 4 ijms-21-03578-f004:**
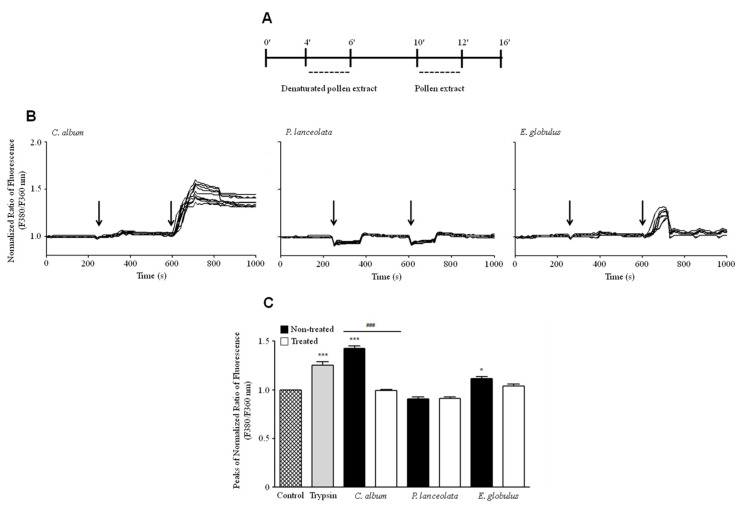
Functional characterization of the effect of pollen proteases from *Chenopodium album* (0.26 ± 0.06 mg/ml), *Plantago lanceolata* (0.18 ± 0.03 mg/ml) and *Eucalyptus globulus* (0.75 ± 0.18 mg/ml) on Calu-3 cells. (**A**) Calu-3 cultures were incubated for 240 s with Krebs solution and stimulated for 120 s with denatured pollen diffusates followed by 120 s incubation with pollen diffusates. (**B**) [Ca2+]_i_ variations in Calu-3 cells after exposure to *C. album* (left graph), *P. lanceolata* (middle graph) and *E. globulus* (right graph). Arrows in each panel indicate the time of addition of the pollen diffusate. The fluorescence profiles are representative of 10 cells. (**C**) The bar graph shows the maximal changes in [Ca^2+^]_i_ after stimulation with each pollen diffusate. Changes in [Ca^2+^]_i_ are depicted by the F*340*/F*380* fluorescence ratios of Fura-2-loaded Calu-3 cells. The black bars correspond to each pollen diffusate, while the white bars correspond to denatured pollen diffusates (95 °C for 30 min). The positive control for this assay was a 0.25% trypsin solution. The data are expressed as the mean ±SEM values of at least 30 representative cell traces using Bonferroni’s post-test: *******
*p* < 0.001; *****
*p* < 0.05 (*n* = 4) in comparison to the control condition and for comparisons between treated and untreated conditions *^###^ p* < 0.001 (*n* = 4).

**Figure 5 ijms-21-03578-f005:**
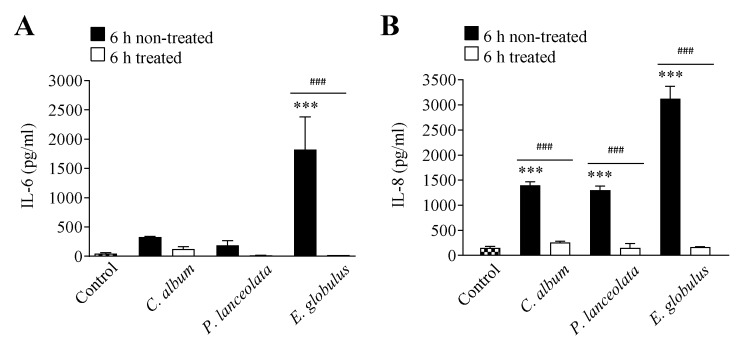
Effect of pollen proteases from *Chenopodium album* (0.26 ± 0.06 mg/ml), *Plantago lanceolata* (0.18 ± 0.03 mg/ml) and *Eucalyptus globulus* (0.75 ± 0.18 mg/ml) on IL-6 (**A**) and IL-8 (**B**) release. The black bars correspond to stimulation with each pollen diffusate, while the white bars correspond to the denatured pollen diffusates (95 °C for 30 min). Cells incubated with culture medium were used as the control condition for this assay. The data are expressed as the mean ± SEM values and were analyzed statistically using Bonferroni´s post-test: *** *p* < 0.001 (*n* = 3) in comparison to the control condition, and *^###^ p* < 0.001 for comparisons between treated and untreated conditions (*n* = 3).

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
