# Peer review of "Pollen Proteases Play Multiple Roles in Allergic Disorders"

_ijms, 2020, doi:10.3390/ijms21103578_

Round 1

Reviewer 1 Report

IJMS 799465

This manuscript focuses on the proteins with enzymatic activity in three different previously underreported species. The authors have attempted to demonstrate that the three different species of pollen elicit a different repertoire of different classes of proteases. Moreover, it was demonstrated that these proteases is capable of degrading the tight junction barrier in epithelial cells using a specific human epithelial cell line. In addition, the specific pathways of this effect was elucidated using immunoblotting to specific proteins involved in tight junction integrity. Furthermore, it was shown that these proteases are able to activate the PAR receptors, which is a known IgE independent pathway to an allergic response. Finally, the authors demonstrate the release of pro-inflammatory cytokines such as IL6 and IL8. The manuscript is well written with logical flow of ideas. Below are my comments;

Major comments:

  • No mention of the allergenic (IgE binding) proteins from these three pollen species have been made. This is important to give the reader a complete picture of the real allergenicity of these pollen species, and not just the ones with proteolytic activity. By extension, it would be good to have a brief comparison with the most frequently reported and diagnosed pollen species in Europe.
  • The introduction speaks in detail the demography of these pollen species, leaving little room for introducing the problem statement, and the objectives of this study. Would be beneficial to discuss more of previous studies on the topic of proteases, epithelial barrier disruption due to allergens, and PAR receptors that play a role in allergic diseases.
  • The introduction and results make abundant use of abbreviations without any mention of the acronym. Please give the full acronym for the first times its used.
  • The diffusates generated form all the three pollen species had different protein concentrations. This concentration was not normalized to an equal value for direct comparison of the results. This leads to the doubt that the comparison of proteolytic activity, effect on barrier junction and PAR activation might be biased. Please provide a justification to not normalizing the protein concentrations to an equal level.

Minor comments:

  • Add a reference to the statement in line 42 , line 50 and line 60
  • Line 59 – change to “widely expressed”
  • Section 2.2 – was the disruption of epithelial barrier tested in a dose-dependent manner?
  • Line 135-136 – Please provide references or justification as to why these specific targets were chosen (E-cadherin, claudin-1, occluding, and ZO1)
  • Section 2.3 – Is Calcium ions an indicator of PAR-2 activation? Any other markers that could have been tested?
  • Section 2.3 – Heat inactivation was linked to denaturation of proteases and subsequently loss of activation of PAR2 and release of cytokines. Could this also be possible that there are heat-stable IgE binding allergens that could have played a role, in these three pollen species?
  • Figure 6 – Any specific reason why E. globulus is able to induce such high levels of IL6 and IL8 release?
  • Line 223 – Epithelial damage “has” been ….

Author Response

Thank you for the review on our manuscript entitled "Pollen proteases play multiple roles in allergic disorders” and the invitation to submit a revised manuscript. As detailed below, we have now modified the manuscript (that follows attached) in response to the concerns of reviewer 1 and 2. With these changes we hope that the manuscript can be accepted for publication in the International Journal of Molecular Sciences.

All the best.

Reviewer #1:

This manuscript focuses on the proteins with enzymatic activity in three different previously underreported species. The authors have attempted to demonstrate that the three different species of pollen elicit a different repertoire of different classes of proteases. Moreover, it was demonstrated that these proteases is capable of degrading the tight junction barrier in epithelial cells using a specific human epithelial cell line. In addition, the specific pathways of this effect was elucidated using immunoblotting to specific proteins involved in tight junction integrity. Furthermore, it was shown that these proteases are able to activate the PAR receptors, which is a known IgE independent pathway to an allergic response. Finally, the authors demonstrate the release of pro-inflammatory cytokines such as IL6 and IL8. The manuscript is well written with logical flow of ideas. Below are my comments;

Response: We thank the reviewer for this positive feedback

Major comments:

No mention of the allergenic (IgE binding) proteins from these three pollen species have been made. This is important to give the reader a complete picture of the real allergenicity of these pollen species, and not just the ones with proteolytic activity. By extension, it would be good to have a brief comparison with the most frequently reported and diagnosed pollen species in Europe.

Response: We thank the reviewer for pointing this out. We have developed these points in the introduction and added new references to the manuscript. All changes have been tracked and can be read on Page 2.

The introduction speaks in detail the demography of these pollen species, leaving little room for introducing the problem statement, and the objectives of this study. Would be beneficial to discuss more of previous studies on the topic of proteases, epithelial barrier disruption due to allergens, and PAR receptors that play a role in allergic diseases.

Response: We thank the reviewer for this suggestion. We have added a new sentence which now reads ”We previously showed that the proteolytic activity of pollen diffusates from Olea europea, Dactylis glomerata, Cupressus sempervirens and Pinus sylvestris compromise the airway epithelial barrier, an effect that is reversed by serine proteases specific inhibitors [9].” on page 2, lines 43-46.

The introduction and results make abundant use of abbreviations without any mention of the acronym. Please give the full acronym for the first times its used.

Response: We have corrected this throughout the document.

The diffusates generated form all the three pollen species had different protein concentrations. This concentration was not normalized to an equal value for direct comparison of the results. This leads to the doubt that the comparison of proteolytic activity, effect on barrier junction and PAR activation might be biased. Please provide a justification to not normalizing the protein concentrations to an equal level.

Response: We have added a new sentence on Page 3, lines 96-99 to elucidate this question which reads: ”Pollen grains were hydrated in vitro to mimic the process that occurs in the respiratory airways when an individual is exposed to pollen inhalation. We used the same initial proportion of pollen mass per volume of buffer to prepare the three diffusates.”. Also in the discussion,  Page 8, lines 242-245 we mention ” In this work, the concentration of protein released by an equal mass of pollen ranged from 0.18 to 0.75 mg/ml. This variation in protein concentration among the analyzed pollens was taken into account in all experiments.”

Minor comments:

Add a reference to the statement in line 42, line 50 and line 60

Response: References have been added to the document.

Line 59 – change to “widely expressed”

Response: We have now made this correction.

Section 2.2 – was the disruption of epithelial barrier tested in a dose-dependent manner?

Response: We have not had the opportunity to do such study but we thank the reviewer for pointing this out as it may be useful for future consideration.

Line 135-136 – Please provide references or justification as to why these specific targets were chosen (E-cadherin, claudin-1, occluding, and ZO1)

Response: We have added new references on Page 5, line 158 where is now reads: ” To understand the mechanism that leads to an increase in transepithelial permeability and cell detachment, potential protein targets for disruption [28,35,36] were identified using immunoblot (Figure 3) and immunocytochemistry” that justify why these specific targets were chosen.

Section 2.3 – Is Calcium ions an indicator of PAR-2 activation? Any other markers that could have been tested?

Response: Mentioned in the discussion on Page 9, lines 277-280 where it reads: ” Moreover, previous studies demonstrated that the hydrolytic activation of PAR-2 leads to phospholipase Cβ activation via the Gq/11α protein, which is a common pathway for G protein-coupled receptors. The activation of this pathway induces the production of inositol triphosphate, followed by Ca2+ mobilization [49,50].”

Section 2.3 – Heat inactivation was linked to denaturation of proteases and subsequently loss of activation of PAR2 and release of cytokines. Could this also be possible that there are heat-stable IgE binding allergens that could have played a role, in these three pollen species?

Response: In both experiments were heat was used to induce denaturation the observed effects were no longer visible (activation of PAR2 and cytokine release). Suggesting that the effects are related to the proteases.

Figure 6 – Any specific reason why E. globulus is able to induce such high levels of IL6 and IL8 release?

Response: Most likely due to the fact of being the pollen with larger amounts of protein released upon hydration and greater specificity in its proteolic activity profile.

Line 223 – Epithelial damage “has” been ….

Response: We thank the reviewer for pointing this out. Change has been made.

Reviewer 2 Report

The manuscript submitted for review is a very thorough presentation of the characteristics of the proteolytic activity of 3 aeroallergens: C. album, P. lanceolate and E. globulus. In my opinion, the article enriches medical knowledge about relevant allergic pollens.

I have no rare objections to work: it is structured correctly, the information is presented comprehensively.

I think that the level of work would increase the presentation of conclusions. The authors may perhaps try to write a few or one sentence inferences from the results presented.

The study concerns on 3 aeroallergens occurring in the Mediterranean region, which is why it will be of interest mainly to researchers from this area. This, of course, is not an accusation, but it indicates a limited interest in this article.

Despite my little tips, the facts presented in the article help to understand the complicated mechanism of allergy pathophysiology.

Author Response

Thank you for the review on our manuscript entitled "Pollen proteases play multiple roles in allergic disorders” and the invitation to submit a revised manuscript. As detailed below, we have now modified the manuscript (which follows attached) in response to the concerns of reviewer 1 and 2. With these changes we hope that the manuscript can be accepted for publication in the International Journal of Molecular Sciences.

All the best.

Reviewer #2

The manuscript submitted for review is a very thorough presentation of the characteristics of the proteolytic activity of 3 aeroallergens: C. album, P. lanceolate and E. globulus. In my opinion, the article enriches medical knowledge about relevant allergic pollens. I have no rare objections to work: it is structured correctly, the information is presented comprehensively. I think that the level of work would increase the presentation of conclusions. The authors may perhaps try to write a few or one sentence inferences from the results presented. The study concerns on 3 aeroallergens occurring in the Mediterranean region, which is why it will be of interest mainly to researchers from this area. This, of course, is not an accusation, but it indicates a limited interest in this article. Despite my little tips, the facts presented in the article help to understand the complicated mechanism of allergy pathophysiology.

Response: We thank the reviewer for this positive feedback. A conclusion has been added to the manuscript summarizing the main findings which reads "This work demonstrated for the first time that pollen diffusates from C. album, P. lanceolata and E. globulus release proteases with a significant in vitro effect on the paracellular permeability of epithelial cells and induced major degradation of cell junction proteins: E-cadherin, claudin-1, occludin and ZO-1. Moreover, our data suggests that the loss of effectiveness of the epithelium barrier may also occur via activation of PAR-2 followed by an immune response characterized by an increase in the release of IL-6 and IL-8." on Page 12, lines 399-404.